# Factors Affecting the Efficiency of In Vitro Embryo Production in Prepubertal Mediterranean Water Buffalo

**DOI:** 10.3390/ani12243549

**Published:** 2022-12-15

**Authors:** Luke Currin, Hernan Baldassarre, Mariana Priotto de Macedo, Werner Giehl Glanzner, Karina Gutierrez, Katerina Lazaris, Vanessa Guay, María Elena Carrillo Herrera, Zigomar da Silva, Caitlin Brown, Erin Joron, Ron Herron, Vilceu Bordignon

**Affiliations:** 1Department of Animal Science, McGill University, Sainte-Anne-de-Bellevue, QC H9X 3V9, Canada; 2Laboratory of Biotechnology and Animal Reproduction—BioRep, Federal University of Santa Maria, Santa Maria 97105-900, RS, Brazil; 3Ontario Water Buffalo Company, Stirling, ON K0K 3E0, Canada

**Keywords:** seasonality, age, individual variation, bull effect, vitrification, LOPU-IVEP, oocyte, embryo transfer

## Abstract

**Simple Summary:**

Embryos can be produced from prepubertal animals using laparoscopic ovum pickup combined with in vitro embryo production technologies. However, due to their young age and unique reproductive physiology, there is currently very limited knowledge about what factors may affect the success of these programs in Mediterranean water buffalo. Here, we assessed how results are affected by season, age, individual variation and choice of sire used for fertilization. Specifically, we found that season and age only had limited impacts, but there were large variations between individual donors and sires. In addition, we assessed what factors can affect pregnancy rates after transferring the resulting embryos into recipient animals and found that regardless of number of embryos transferred, fresh or vitrified, all variables assessed were compatible with the establishment of pregnancies.

**Abstract:**

Embryos from prepubertal water buffalo can be produced using laparoscopic ovum pickup (LOPU) and in vitro embryo production (IVEP). However, to date, it is unclear what factors and environmental conditions can affect LOPU-IVEP efficiency in prepubertal animals, especially buffalo. In this study, we explored the impact of season, age and individual variation among female donor animals, as well as the effect of the sire used for in vitro fertilization. Donor animals between 2 and 6 months of age were stimulated using gonadotropins prior to LOPU, which was performed at two-week intervals. Following in vitro maturation and fertilization, the resulting embryos were then cultured to the blastocyst stage until they were either vitrified or transferred into recipient animals. The number of follicles available for aspiration and embryo development rates was stable throughout the year. As animals became older, there was a slight trend for fewer COCs recovered from LOPU and better embryo development. There was a large individual variation in both ovarian response and the developmental competence of oocytes among donors. The bull used for fertilization also had a significant impact on embryo development. Upon embryo transfer, pregnancy rates were not affected by the number of embryos transferred per recipient. The best pregnancy rates were achieved when transferring blastocysts, compared to compact morula or hatched blastocysts. Finally, vitrification had no effect on pregnancy rate compared to fresh embryos.

## 1. Introduction

Famous for their mozzarella cheese, Mediterranean water buffalo (*Bubalus bubalis*) are an important livestock species, for both meat and dairy production worldwide. In the developed world, water buffalo milk is prized for its high protein and fat content [1], while, in developing countries, their meat offers an alternative and affordable healthy protein source [2,3]. With a sharp rise in demand for buffalo products in recent years, a key constraint for sustainable and efficient buffalo production is their unique and sometimes challenging reproductive biology [4]. Therefore, adept implementation and use of assisted reproductive technologies (ARTs) in buffalo is crucial for the long-term viability and profitability of raising this species [5].

Despite a wide array of ARTs used commercially in dairy cattle, implementation in water buffalo has lagged. Compared to dairy cattle, adult water buffalo respond poorly to multiple-ovulation and embryo transfer (MOET) programs with very few viable embryos being recovered upon flushing [5,6,7]. Hence, attention instead turned to ultrasound-guided transvaginal ovum pickup (OPU) followed by in vitro embryo production (IVEP). However, this has also yielded poor results compared to dairy cattle, with an average yield of < 2 transferable embryos per OPU in adult donors [8,9,10]. To further compound this problem, the onset of puberty and age of sexual maturity in water buffalo is typically delayed, making rates of genetic gain very slow [4]. Although the age of first calving is affected by various factors including nutrition [11], the interval between generations is a major factor impeding rates of genetic gain. Hence, a viable alternative is recovering cumulus-oocyte complexes (COCs) from prepubertal animals using laparoscopic ovum pickup (LOPU) combined with IVEP to produce embryos from elite donor animals at a very young age [12,13,14]. As LOPU can safely be repeated at regular intervals on animals as young as two months of age [14], prepubertal LOPU-IVEP programs represent an exciting biotechnology that dramatically decreases the generation interval and has been used in multiple ruminant species including buffalo [5,15], cattle [16,17,18], goat [19,20] and sheep [21,22]. For example, although embryo development rates from prepubertal animals have historically remained low compared adult animals [12], recent studies in our laboratory produced an average of 2 to 3 transferrable embryos per donor per LOPU [23], which is comparable or even slightly better than averages for OPU-IVEP/MOET programs in adult buffalo [10,24]. Following embryo transfer into adult recipient animals, these embryos produced healthy, live calves proving the in vitro embryos produced from prepubertal animals are fully viable [5,13]. However, to date, it is unclear what factors affect the efficiency of these programs.

One distinct aspect of reproductive physiology in water buffalo compared with dairy cattle is their seasonality. Adult water buffalo are known to be photoperiod-sensitive, with season being an important factor affecting the success of breeding programs [25,26]. Seasonal animals can be broadly divided into short-day breeders, which are most fertile in the autumn and winter, and long-day breeders, which are most fertile in the spring and summer [27]. Like sheep, goat, and deer, water buffalo are short-day breeders with fertility increasing as the daylength draws shorter in the autumn and winter [25,28]. As such, heat detection, artificial insemination, embryonic survival and pregnancy rates all improve during the autumn months [29,30,31,32]. These effects also carry over into embryo development rates of OPU-derived oocytes during IVEP as well, with cleavage and blastocyst rates significantly affected by season [8,25]. Photoperiod-sensitivity is driven by the hypothalamic-pituitary-ovarian (HPO) axis with melatonin, which is produced by the pineal gland in response to darkness, modulating GnRH production, and consequently, gonadotropin secretion in accordance to daylength [33,34]. However, studies investigating the impact that season may play on very young prepubertal buffalo are lacking and needed, since, unlike adult animals, prepubertal animals have an under-developed and immature HPO axis [35]. Therefore, it is plausible that prepubertal buffalo may not respond to season in the same way as adult animals, although to our knowledge no one has investigated this.

Another key aspect affecting the success of prepubertal LOPU-IVEP is individual variation among animals. Multiple studies have shown that some individuals consistently respond very well to gonadotropin stimulation and produce many COCs and embryos, while other animals produce very few—if any, despite being housed in the same conditions [12,14,18,24]. This has made choosing which donor animals to select for LOPU-IVEP programs challenging. In a similar manner to the large variation observed among donor females, the choice of male for use in IVF is a key consideration in buffalo IVEP, as different bulls may have different fertility, and consequently, embryo development rates [5,36]. For example, working with adult Murrah buffalo, a study comparing 8 different bulls for IVF resulted in embryo development rates varying between 6.4 and 37.7% [5]. In fact, it has been suggested that only 10% of bulls are suitable for buffalo IVF [5,24]. As oocytes sourced from prepubertal females are known to have multiple molecular and organelle differences compared to oocytes from adult animals [17,37,38,39], it is possible that the bull used for IVF with prepubertal oocytes may have an even larger influence. Although the polyspermy rate in prepubertal oocytes has been shown to be abnormally high [14,18], to date, we are unaware of any authors investigating the effect of bull on oocytes from prepubertal buffalo.

In addition to season and individual variation, the age of the donor at the time of LOPU has been shown to impact results in dairy cattle. Compared to their adult counterparts, prepubertal animals have a large ovarian reserve and often respond very well to gonadotropin stimulation, yielding many follicles available for aspiration [40,41]. However, despite yielding many COCs, in vitro embryo production rates are low, with similar trends shown across multiple species including buffalo [5], cattle [17,42], goat [20], sheep [21,22] and pigs [43,44]. On this note, age has been shown to affect LOPU-IVEP success in other species [45,46]. For example, in Holstein cattle, we showed that as animals became older, although the number of COCs recovered from LOPU gradually decreased, the competence of those oocytes grew, with embryo development rates improving [16]. Similar trends have been noted in goats [19]. Finally, it is important to understand the viability of embryos once they are transferred into recipient animals. Although multiple authors have investigated embryo development rates, fewer have assessed what factors may influence pregnancy rates following embryo transfer. Even though pregnancy rates established using embryos produced from prepubertal animals have generally been comparable with those using regular embryos in recent years [14,15,16,47], there is only limited information available about what factors can affect these rates.

In this retrospective study, working with buffalo heifer calves between 2 and 6 months of age, we assessed the impact of season, donor age, and individual variation in both the donors and the males on the efficiency of LOPU-IVEP. Specifically, we assessed the ovarian response based on the number of follicles aspirated, as well as the number of COCs recovered following LOPU. We also assessed cleavage and in vitro development to the blastocyst stage. Finally, we assessed factors that can affect the pregnancy rate following embryo transfer into recipient females, including season, the number of embryos transferred, as well as whether the prepubertal-derived embryos were transferred fresh or vitrified.

## 2. Materials and Methods

### 2.1. Chemicals and Reagents

Unless otherwise indicated, all chemicals and reagents were purchased from the Sigma Chemical Company (Millipore Sigma; Oakville, ON, Canada).

### 2.2. Animals

A total of 256 LOPUs on 55 Mediterranean water buffalo heifer calves between the ages of 2 and 6 months old were performed in different seasons between the summer of 2017 and spring of 2022, every two weeks up to a maximum of seven times per animal. Donor animals were housed in an indoor barn at Macdonald Campus of McGill University, located in Sainte-Anne-de-Bellevue, QC, Canada (45°25′30″ N, 73°57′55″ W). The barn had large windows with lots of natural light; during the summer, the barn was kept as cool as possible with large fans, while during the winter it was heated to 15 °C. Animals were weaned and fed good quality second-cut hay and water offered ad-libitum, as well as a grain concentrate (Optivia^®^, Shur-Gain, Brossard, QC, Canada) twice daily according to their body weight. Recipient adult animals were housed in a free-stall barn with access to the outdoors at the Ontario Water Buffalo Company farm in Stirling, ON, Canada (44°20′16″ N, 77°34′47″ W).

### 2.3. Ovarian Stimulation

Animals were treated intramuscularly with a combination of 100–140 mg of FSH (Folltropin-V^®^, Vétoquinol, Lavaltrie, QC, Canada) with or without 300–600 IU eCG (Novormon^®^ 5000, Partnar Animal Health Inc., Ilderton, ON, Canada) starting 72 to 120 h prior to LOPU. FSH was diluted in either 0.9% saline solution, given at 12-h intervals, or hyaluronan (MAP-5^®^, Vétoquinol), given in a single injection or two injections 48-h apart. All animals received two vaginal progesterone implants (330 mg each) in the form of two small ruminant CIDRs (EAZI-Breed^TM^ CIDR^®^ 330, Zoetis Canada Inc., Kirkland, QC, Canada) inserted 120 h prior to LOPU. Results regarding the effect of gonadotropin treatment were demonstrated in a previous publication from our group [23]. Since no significant differences among each parameter assessed was found, for the purposes of this retrospective study, data was pooled.

### 2.4. Laparoscopic Ovum Pickup (LOPU)

Prior to LOPU, animals were fasted of hay for 36 h, concentrate for 24 h, and water for 18 h. Anesthesia was induced using 0.05 mg/KBW xylazine (Xylamax, Bimeda Canada, Cambridge, ON, Canada), 2 mg/KBW ketamine (Ketalean, Bimeda Canada) and 0.1 mg/KBW diazepam (Sandoz, Boucherville, QC, Canada) administered intravenously. Animals were then intubated, attached to a mechanical ventilator, and maintained under general anesthesia using 2% isoflurane USP (Fresenius Kabi, Toronto, ON, Canada). The laparoscopic ovum pickup technique has been described previously [48]. Briefly, once fully anesthetized, animals were moved into the Trendelenburg position in a cradle. Follicles were aspirated under laparoscopic observation using a 20G needle mounted on an acrylic pipette which was connected to a vacuum pump. The vacuum pump was adjusted to 50 mmHg of pressure at the pump and 60 drops of media reaching the collection tube per minute using a flow-valve on the vacuum-line tubing. The aspiration medium was composed of HEPES-buffered TCM-199 (Life Technologies, Burlington, ON, Canada) supplemented with 2 mg/mL BSA, 2 μL/mL gentamicin and 0.05 mg/mL heparin. Following aspiration, the ovarian surface was rinsed with 0.9% saline and the incisions were closed using surgical glue. Finally, each animal had their CIDR removed and received 1 mL/10 KBW long acting oxytetracycline 200 mg/mL (Oxymycine^®^ LA, Zoetis, Kirkland, QC, Canada), 1.5 mL/50 KBW ketoprofen (Anafen^®^ 100 mg/mL, Merial Canada Inc., Baie d’Urfé, QC, Canada) and monitored until they had fully recovered.

### 2.5. Washing and Grading of COCs

Following transfer of the collection tube to the laboratory, COCs were recovered and washed in manipulation media composed of HEPES-buffered TCM-199 (Life Technologies) supplemented with 2 mg/mL BSA, 2 μL/mL gentamicin. Under a stereomicroscope, COCs were graded based on their morphology as either grade 1 (>3 layers of compact cumulus cells and evenly granulated ooplasm), grade 2 (1–3 layers of compact cumulus cells and evenly granulated ooplasm), grade 3 (absent cumulus oophorus), or grade 4 (expanded cumulus oophorus, heterogeneous ooplasm or degenerated). Grade 1 and 2 COCs were selected as usable and transferred into in vitro maturation.

### 2.6. In Vitro Embryo Production

Unless otherwise indicated, IVEP was conducted in commercial bovine media (L’Alliance Boviteq, St. Hyacinthe, QC, Canada).

#### 2.6.1. In Vitro Maturation (IVM)

After grading and selection, usable COCs were placed into 50 μL drops of maturation media under mineral oil (Fisher Scientific, Ottawa, ON, Canada) in groups of around 10 after being washed. Maturation media consisted of TCM-199, 100 μg/mL cysteine, 5 UI/mL hCG (Chorulon^®^; Merck Animal Health, Kirkland, QC, Canada), 10 μg/mL FSH (Folltropin-V^®^), 1 μg/mL 17 *β* Estradiol, 0.2 mM pyruvate, 2 mM L-carnitine, 5 μL/mL insulin-transferrin-selenium (ITS), 3 μL/mL stock FLI, 10 ng/mL epidermal growth factor (EGF; Life Technologies), 50 μg/mL gentamicin, 10% *v/v* fetal bovine serum (FBS). Stock FLI contained 20 ng/mL recombinant human leukemia inhibitory factor (LIF; Peprotech, Cranbury, NJ, USA), 20 ng/mL recombinant human insulin-like growth factor 1 (IGF1; Peprotech) and 40 ng/mL recombinant human fibroblast growth factor 2 (FGF2; Gold Biotechnology, St. Louis, MO, USA). COCs were incubated at 38.5 °C in 5% CO_2_ and 95% air for around 24 h.

#### 2.6.2. In Vitro Fertilization (IVF)

IVF was conducted in 45 μL droplets of FERT medium containing 1mM penicillamine, 1 mM hypotaurine, 250 mM epinephrine and 10 μg/mL heparin plated under mineral oil. Following IVM, COCs were washed in FERT medium and placed into the droplets in groups of around 5. A straw of frozen water buffalo semen of known fertility was thawed in a 37 °C water bath for 1 min and filtered through a discontinuous gradient (45% over 90%) of Bovi-Pure^®^ (Nidacon Laboraties AB, Göthenborg, Sweden) by centrifugation at 600× *g* for 10 min. The supernatant was discarded, and the pellet resuspended in gradient medium. The sperm was then centrifuged a second time (5 min at 600× *g*) before the pellet was finally resuspended in 1 mL FERT medium. Sperm motility was assessed and then counted in a haemocytometer. The concentration was then adjusted and 5 μL was added to each 45 μL droplet containing COCs for a final concentration of between 2–4 × 10^6^ motile sperm per mL, as per the experiment described below. COCs and sperm were co-incubated overnight for around 18 h at 38.5 °C in 5% CO_2_ and 95% air.

#### 2.6.3. In Vitro Culture (IVC)

Presumptive zygotes were washed to remove cumulus cells and sperm stuck to the zona pellucida following IVF before being placed in 30 μL droplets of culture media supplemented with 2mM L-Carnitine in groups of around 10 zygotes, plated under mineral oil and incubated at 38.5 °C with 100% humidity in an atmosphere of 5% CO_2_, 5% O_2_ and 90% N_2_. After 4 and 6 days of culture, embryos were transferred into new droplets. Cleavage was assessed 96 h after IVF and embryo development rates were assessed after 7 days. Excellent grade, compact morulae and blastocysts were determined to be of transferrable/cryopreservation quality based on morphology.

### 2.7. Embryo Vitrification and Thawing

Embryos were vitrified using a protocol adapted from a previously described method [49]. Embryos were selected for vitrification in groups of 1 to 3. Vitrification was performed in a four-well NUNC plate on a warm plate with medium pre-warmed to 37 °C. In wells 1 and 2, embryos were washed in 800 μL holding medium composed of TCM-199 HEPES-buffered medium (Life Technologies, Burlington, ON, Canada) supplemented with 20% fetal bovine serum (FBS). Embryos were then moved into well 3 containing 1 mL of VS1 medium composed of holding medium with 7.5% ethylene glycol and 7.5% dimethyl sulfoxide (DMSO) for three minutes. Embryos were then moved into the fourth well containing 1 mL VS2 medium composed of holding medium supplemented with 670 mM sucrose containing 16.5% ethylene glycol and 16.5% DMSO for 1 min. Embryos were then loaded onto a properly labelled vitrification straw approximately 2 mm from the end in the smallest volume possible. Vitrification straws were made from 0.5 cc semen straws with a 30° bevel cut into one end. The straw containing the embryos was then immediately plunged into liquid nitrogen and kept submerged for at least 30 s. Straws were then covered with a sheath and loaded into goblets before being arranged into canes for storage.

Embryos were thawed using a four-well NUNC plate on a warm plate with media pre-warmed to 37 °C on the day of embryo transfer. Straws were removed from the liquid nitrogen and the tip containing the embryos was immediately plunged into the first well containing 1200 μL of holding medium supplemented with 333 mM sucrose. Embryos were recovered and moved into the second well containing the same medium and left to incubate for 5 min. Embryos were then moved into the third well containing 1200 μL of holding medium supplemented with 167 mM sucrose and incubated for another 5 min. The embryos were then moved into the final well containing 1200 μL of holding medium and washed thoroughly. Embryos were then washed in transfer medium and loaded into straws ready for transfer to the recipient females.

### 2.8. Embryo Transfer and Pregnancy Detection

Embryos were loaded into embryo transfer straws in the laboratory and transported in a portable incubator at 38.5 °C. When transferring fresh embryos, adult recipient animals were synchronized to be in heat on the same day as LOPU, for embryo transfer to occur eight days later. Twenty-one days before ET, a bovine progesterone implant was inserted (EAZI-BREED^TM^ CIDR^®^ 1380; Zoetis Canada), and it was removed ten days later. Animals received 400 IU eCG (Folligon, Intervet Canada, Kirkland, QC, Canada) and 375 μg cloprostenol (Estrumate^®^, Merck Animal Health, Kirkland, QC, Canada) at the time of CIDR removal. Forty-eight hours later, animals received 50 μg gonadorelin GnRH (Cystorelin^®^, Merial Canada). The day before ET, recipients were assessed for the presence of corpora lutea. One to two embryos were transferred non-surgically into the uterine horn ipsilateral to the corpus luteum. Pregnancy was assessed 30–40 days following ET using trans-rectal ultrasonography.

### 2.9. The Effect of Season

Over the period of five years, LOPU-IVEP was performed in a group of 53 animals up to a maximum of 7 times, for a total of 248 replicates. To assess the impact of season, the date of LOPU was divided into one of four seasons of equal length: spring (March-May; *n* = 88), summer (June-August; *n* = 55), autumn (September-November; *n* = 81) and winter (December-February; *n* = 28). The animal housing facility is located in the St. Lawrence Lowlands, at a 45° latitude, winters are cold and dark, while summers are hot, humid and lengthy in daylight. The average daily meteorological conditions for the study area recorded during the trial are shown in Table 1. For the effect of season on embryo transfer, the date of embryo transfer was considered rather than the date of LOPU. A total of 118 embryo transfers were performed across all four seasons: March-May (*n* = 31), June-August (*n* = 31), September-November (*n* = 23), December-February (*n* = 33).

### 2.10. The Effect of Age

To evaluate the effect of age, LOPU results were assessed on a group of 55 animals, up to a maximum of 7 procedures, for a total of 256 replicates. Animals were pooled into one of three age categories according to their age at the time of LOPU: (A) ≤ 120 days old (*n* = 78); (B) between 120 and 150 days (*n* = 98), and (C) ≥ 150 days old (*n* = 80).

### 2.11. The Effect of Individual Variation

The effect of individual variation was evaluated on a group of 6 animals on which we conducted LOPU a total of 4 times each, for a total of 24 replicates. The same frozen semen batch/bull was used for all replicates.

### 2.12. The Effect of Sire and Semen Concentration

The effect of sire on IVEP was assessed in 8 animals over 3–4 LOPUs, with a total of 5 different bulls tested. To reduce the error between subsequent replicates, for any given LOPU, oocytes from the same donor animal were split and fertilized with semen from two different bulls for donor animals who produced enough COCs. A total of 50 replicates were used in this study: bull A (*n* = 8), bull B (*n* = 8), bull C (*n* = 8), bull D (*n* = 7) and bull E (*n* = 19).

The effect of semen concentration was studied on a group of 8 donor females over 4 LOPUs. In all cases, IVF was performed using a single frozen semen source (bull D) using a concentration of 2, 3 or 4 million motile sperm/mL in the IVF drops. As described previously, for animals that produced enough COCs, multiple concentrations were tested in the same female and LOPU. A total of 35 replicates were used in this study: 2 million (*n* = 11), 3 million (*n* = 8), 4 million (*n* = 16).

### 2.13. Factors Impacting Embryo Transfer

We studied the impact of the number of embryos transferred (1 or 2), the stage of embryo development at the time of transfer (morula, blastocyst or hatched blastocyst) and the type of embryo transferred (vitrified or fresh).

### 2.14. Statistical Analysis

Data was analyzed using the JMP software (SAS Institute Inc., Cary, NC, USA). The normality of data were tested using the Shapiro-Wilk W test and normalized when necessary. For the effect of season and age on LOPU and IVEP results, a one-way ANOVA followed by a Tukey-Kramer HSD test was performed. The effect of individual variation and sire used for IVF was compared using a student’s t-test. Embryo transfer and pregnancy data was compared using a chi-square test in a contingency table. Results are expressed as the average ± the standard error of the mean. Differences were considered statistically significant at the 95% confidence interval (*p* < 0.05).

## 3. Results

### 3.1. The Effect of Season

The number of follicles available for aspiration was similar across all four seasons, varying between 16.0 ± 1.2 and 16.3 ± 1.0 follicles in the spring, autumn and winter, with a slight decrease during the summer months (14.3 ± 1.0; Figure 1A). This led to uniform number of COCs recovered throughout the year, varying between 12.3 ± 0.9 and 13.6 ± 1.0 (Figure 1A). The percentage of COCs that were deemed usable was numerically higher in the autumn and winter (82.1 ± 2.2 and 85.4 ± 3.3%) compared to the spring and summer (77.4 ± 2.1 and 81.3 ± 2.2%). Cleavage rates and embryo development rates, calculated over both oocyte and cleaved, were homogeneous regardless of season (Figure 1B). This led to the production of approximately two embryos per donor per LOPU throughout the year, varying between 1.84 ± 0.3 and 2.22 ± 0.4 (Figure 1C). Upon embryo transfer into adult recipient animals, there was a slight numerical increase in pregnancy rates in the autumn and winter (30.4 and 33.3%), however, because of the small sample size, this difference was not statistically significant compared to the spring and summer (25.8 and 22.6%; Table 2). When nulliparous heifer recipient animals were excluded from the data set, and only primiparous or multiparous cow recipients were considered, the pregnancy rate was numerically lower in the summer months, with only 1/16 transfers resulting in pregnancy (6.25%) compared to the other seasons which varied between 20.0 and 37.5%.

### 3.2. The Effect of Age

As donor animals got older, there was a slight numerical, albeit statistically insignificant (*p* > 0.05), decrease in the number of follicles aspirated from 17.3 ± 1.2 to 15.5 ± 1.1. This led to a similar decline in the number of COCs recovered following LOPU, falling from 14.7 ± 1.1 to 12.9 ± 1.1 (Figure 2A). However, the proportion of COCs that were deemed usable slightly increased with age, increasing from 79.5 ± 1.8 to 81.7 ± 2.3% (*p* > 0.05). Following IVEP, the trends reversed, with cleavage (44.1 ± 4.4 to 52.1 ± 2.8%), embryo/oocyte (20.8 ± 3.6 to 23.5 ± 2.6%) and embryo/cleaved (41.3 ± 5.8 to 43.0 ± 3.9%), all slightly increasing with age, although this was not statistically significant (Figure 2B). Collectively, this resulted in a mean number of between 2 and 3 embryos per donor per LOPU (2.11 ± 0.4 vs. 2.44 ± 0.3 vs. 2.54 ± 0.3; *p* > 0.05; Figure 2C).

### 3.3. The Effect of Individual Variation

There was a large individual variation among donor females, both in ovarian response, and embryo development rates during IVEP. For example, the number of follicles aspirated per female varied from a mean of 5.0 ± 1.6 to 24.5 ± 1.4 per LOPU. Over the course of 4 LOPUs, this resulted in a range from 20 to 98 follicles aspirated per animal. In terms of number of COCs recovered per animal, there was a similar large variation from 4.5 ± 1.0 to 18.0 ± 2.8 per LOPU (Figure 3A) or from 18 to 72 total COCs per animal over 4 LOPUs.

There was also a large variation in embryo development rates among females, despite using the same conditions (bull, semen concentration, capacitation factors, etc.) for all animals during IVEP. For example, cleavage rates varied from 34.9 ± 3.0% to 74.3 ± 15.2% and mean embryo development rates calculated over oocyte varied from 8.0 ± 5.9% to 52.0 ± 9.8% (Figure 3B). This resulted in a mean number of embryos per donor ranging from 0.75 ± 0.3 to 5.5 ± 1.6 (Figure 3C), resulting in a total number of blastocysts per donor ranging from 3 to 22 for the four LOPUs. It is interesting to note that animals who had good ovarian responses did not necessarily result in superior IVEP responses and vice versa.

### 3.4. The Effect of Sire and Semen Concentration

In a similar manner to the large individual variation among donor females, there was a large variation on IVEP results depending on which bull was used for IVF. In the same group of eight female donor calves, we tested five different bulls, labelled A through E (Figure 4A). We used the same concentration for each bull, 3 million motile sperm/mL. Among bulls, there was a large variation in cleavage rate, ranging from 10.1 ± 6.5 to 73.6 ± 4.4%. Due to the small sample size, there were relatively large variations in embryo development rate calculated over cleaved, as evidenced by the error bars. However, the mean varied between 25.0 ± 16.4 and 66.3 ± 14.2%. It is interesting to note that although bull B resulted in the highest cleavage rate, it resulted in the second lowest embryo rate. Collectively, this resulted in an average embryo rate/oocyte varying between 3.9 ± 3.1 and 47.0 ± 6.0%.

Next, we assessed the effect of semen concentration using a single group of donor animals (*n* = 8) and bull D (Figure 4B). Although not statistically significant (*p* > 0.05), compared to 2 and 4 million, inseminating COCs with 3 million motile sperm/mL led to the greatest cleavage (52.2 ± 9.7%) and embryo development rates calculated over both cleaved and oocyte (56.1 ± 13.1 and 29.8 ± 7.9, respectively).

### 3.5. The Effect of Embryo Transfer

There was a limited availability of adult recipient animals, so embryos produced from LOPU-IVEP were either vitrified or transferred directly into recipient animals on day 7 of development. Either one or two embryos were transferred into recipient animals at different stages of development: compact morula, blastocyst or hatched blastocyst (Table 3). Although most of the recipient animals were still pregnant at the time of manuscript submission, to date, the vitrified embryos have resulted in five healthy, live calves born (4 heifers and 1 bull). We believe these are the world’s first live calves born from vitrified embryos produced from a prepubertal LOPU-IVEP program in buffalo.

## 4. Discussion

In this first of its kind study, we showed that embryos from prepubertal buffalo heifers can be produced year-round, despite large individual variations among oocyte donors and sires used for IVF. Furthermore, we showed that embryos produced from prepubertal buffalo oocytes were fully viable after vitrification, producing healthy live young, which may substantially facilitate the technology’s applicability going forward. The efficiency of LOPU-IVEP in prepubertal buffalo has been growing, with specially tailored gonadotropin stimulation and in vitro maturation procedures yielding promising results [23]. Further elucidating the impacts of exogenous factors on LOPU-IVEP programs is crucial for adoption of this technology on a commercial basis in the near future.

In this study we found that the effect of season did not have a large impact on LOPU-IVEP results, obtaining similar follicular and embryo development results year-round. In adult Mediterranean water buffalo, although season was shown to only have a small impact on the number of follicles available for aspiration and COCs recovered during OPU, embryo development rates significantly improved during the autumn months compared to the spring and summer (30.9% vs. 13.3%) [8]. This contrasts with our results, where we obtained embryo development rates around 20% year-round. This could potentially be explained given the immaturity of the HPO axis in prepubertal animals, therefore, without being dependant on, or affected by endogenous gonadotropin production, the administration of exogenous gonadotropin stimulation prior to LOPU resulted in similar responses year-round. While many factors affect the seasonality of buffalo, with both exogenous (e.g., climate, nutrition) and endogenous (e.g., genomic, endocrine) elements playing a role, photoperiod is likely the principal regulator [28]. This is especially true at higher latitudes, where day length is more variable than sub-tropical areas closer to the equator [10,51]. Consequently, the recipient animal was probably the major driver behind the higher pregnancy rates achieved during the autumn and winter (18/56 = 32.1%), compared to the spring and summer (15/62 = 24.2%). This is consistent with previous studies in adult animals showing similar changes in pregnancy rate according to season [52,53]. Our findings further emphasize the role photoperiod has on seasonality, as our results are similar to other studies across the northern hemisphere [54], despite the fact that Canada’s climate involves much milder temperatures during spring and summer compared to other regions of the world (southern Italy for example). It is interesting to note, however, that the physiological rise in melatonin plasma concentrations in response to darkness is lower in buffalo heifers compared to adult cows [55], with seasonal effects on fertility more pronounced in adult cows than in heifers [54,55]. The fact that only 1/16 transfers to adult cows resulted in pregnancy during the summer months seems to reflect this. It is also possible that cows were in a higher degree of metabolic and heat stress during summer due to lactation demands, explaining their higher degree of difficulty to become/remain pregnant in the summer compared with heifers. Although we did not directly measure melatonin production in this study, future research investigating melatonin synthesis across multiple age groups, in both gonadotropin-stimulated, and unstimulated calves remains an interesting avenue for future studies that could shed light on the mechanisms affecting results in younger animals.

Although there was a slight trend in our results, we did not see a large variation in results according to age. Previous results in our lab using Holstein calves of similar ages to buffalo calves used in this study showed that as animals got older, although the number of COCs recovered from LOPU gradually decreased, embryo development rates improved [16]. Since buffalo mature at a far slower pace than Holsteins, reaching puberty at a much later age [4], it is possible the timeframe in this study was too narrow for us to detect a significant effect. However, our results are consistent with previous knowledge that prepubertal animals produce substantially more follicles and COCs compared to adults [40,41,56]. For example, although we saw a slight trend in lower numbers of COCs being recovered as animals became older (14.7 ± 1.1 to 12.9 ± 1.1), this is still substantially higher than an average of fewer than three COCs recovered per ovary from slaughterhouse-derived embryos [57,58,59,60]. This is consistent with other research which following LOPU in prepubertal, and OPU in adult buffalo, recovered an average of 10.9 ± 3.3 and 5.8 ± 1.3 COCs, respectively [47]. One drawback of our experimental design was that since LOPU was repeated at two-week intervals, we were unable to discriminate between the effect of age and repeated gonadotropin stimulation. It is possible that animals began to build an immunogenic response to gonadotropins after repeated gonadotropin stimulation, and consequently, produced fewer follicles. One way to avoid this potential source of error would be to only assess each animal once; however, a major source of error would then be introduced with individual variation among animals. Nonetheless, previous research in adult goats has shown that repeated LOPU does not have an impact on the number of follicles available for aspiration [61].

We showed that there was a large variation among donor animals, which is consistent with previous studies, in both prepubertal [14], and adult buffalo [24]. This is a potential challenge for choosing which animals are best suited to be enrolled in LOPU-IVEP programs. For example, in this multi-year study, considering the sum of 4 LOPUs per donor, our best animal produced 29 transferrable embryos, while our poorest animal produced zero. In adult animals, there is a correlation between antral follicle count (AFC) and the number of blastocysts produced in subsequent OPU-IVEP sessions [24]. However, because prepubertal animals are too small for rectal ultrasound imaging, this metric is less useful in young animals. Concentrations of anti-Müllerian hormone (AMH) have been proposed as a reliable marker for gonadotropin-stimulation response in adult buffalo [47,62]. Although it is unclear whether this remains true in prepubertal buffalo, studies in *Bos taurus* and *indicus* calves have shown it to be a valuable and accurate tool for identifying animals with high AFC [63].

Despite the fact that all bulls tested in this study were proven to be fertile when used for artificial insemination, we found a large variation in IVF results (3.9 to 47.0% embryo/oocyte rate) among five bulls. This is consistent with previous research that found only around 10% of bulls were suitable for IVF in buffalo [5,24]. To further complicate this issue, it has also been shown that there is a large variation in fertilization kinetics among bulls in buffalo, with different males requiring different co-incubation lengths to successfully penetrate oocytes [64]. Although in cattle there is a correlation between AI and IVF fertility [65], the large variation among bulls in buffalo IVF suggests different bulls require different conditions in vitro. Although we did not test different parameters in this study, previous studies in buffalo found that different bulls expressed different abilities to capacitate in vitro regardless of treatment, and that different bulls responded differently to different capacitation factors [66] and heparin concentrations [67]. This further suggests that various bulls may require capacitation factors specifically tailored to their needs. Another possibility is that different bulls may require different sperm concentrations in order to perform optimally. Although we did not test this specifically, we found that 3 × 10^6^ motile sperm/mL to be optimal for bull D. However, this is higher than the industry-standard concentration of 2 × 10^6^ [67,68], further lending evidence to the theory that different bulls may require specific IVF conditions.

We observed that pregnancy rate was numerically higher but not statistically different when two embryos were transferred together compared to one embryo (29.3 vs. 22.2%). Since we had limited access to recipients and we predicted in vitro produced embryos from prepubertal buffalo would have a rather low survival rate, we thought transferring embryos in pairs would allow more calves on the ground with a rather low incidence of twins (3/29 pregnancies = 10.3%). Since this did not work the way we expected, the recommendation is to transfer embryos individually, as they resulted in a better pregnancy rate per embryo (22.2 vs. 16.2%). The results also reaffirm the importance of the recipients in the overall success of an embryo transfer program [53]. On the embryo side, one factor that does appear to have an impact on results is development stage at the time of transfer. Indeed, we observed that transferring blastocysts led to the highest pregnancy rates (47.4%) compared to morula (20.0%) or hatched blastocysts (26.7%).

One particularly noteworthy finding of our study was that vitrified embryos were able to establish pregnancies at the same rate compared to embryos transferred fresh (28.6 vs. 28.7%). Further opening the possibility of embryo sale and export when prepubertal LOPU-IVEP is used commercially. Embryo quality is known to be a major factor affecting cryotolerance [69,70], further emphasizing the quality of embryos that were obtained in our study. Indeed, although most recipients were still pregnant at the time of manuscript submission, vitrified embryos did yield healthy, live calves proving the embryos we produced were fully viable. To date, a total of five calves have been born from vitrified embryos, four heifers and one bull. One factor that may partially explain this is that oocytes from prepubertal donor animals appear to be less sensitive to cryoprotectants than adult oocytes [71]. However, following vitrification of caprine blastocysts, re-expansion rates were lower in those produced from prepubertal donors compared to adult goats [20]. In adult OPU-derived embryos, pregnancy rates between fresh and vitrified embryos were 43.4 and 37.1% after 30 days and 41.7 and 31.4% after 60 days, respectively [9]. However, similar to our results, other researchers found no significant differences in pregnancy rates between direct transfer and vitrified buffalo embryos [47]. One particular characteristic of buffalo embryos making vitrification difficult is their high lipid content [68]. To address this, we supplemented our IVEP medium with L-carnitine, which has been shown to promote lipid metabolism [72,73], provide antioxidative support and improve post-thaw survivability [74,75,76,77].

## 5. Conclusions

Findings from this study revealed that season does not appear to affect LOPU-IVEP efficiency in prepubertal buffalo, indicating that LOPU-IVEP can be performed year-round with negligible effects on efficiency. In addition, pregnancy rates were slightly better in the autumn and winter, presumably due to the seasonality of recipient animals, and were not affected by embryo vitrification compared to fresh embryo transfer. As such, it may be beneficial to vitrify embryos produced during the spring and summer from particularly valuable donor animals, for transfer in the autumn and winter to further maximize the number of calves born. Furthermore, we found a large individual variation in both donor females and the male used for IVF, but donor age only had a small impact on results, suggesting that prospective donor animals and the bull used for IVF should be carefully selected.

## Figures and Tables

**Figure 1 animals-12-03549-f001:**
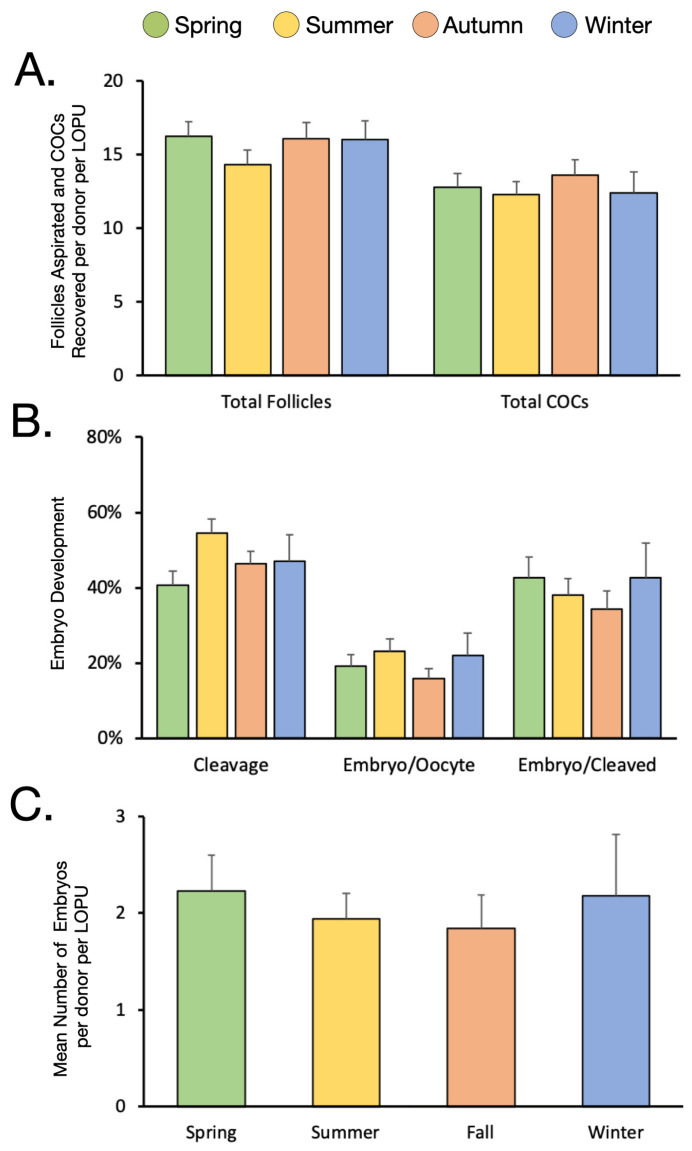
The effect of season on the number of follicles and COCs recovered during LOPU as well as in vitro embryo production rates. (**A**): LOPU results by season. (**B**): Embryo development rates by season. (**C**): The mean number of embryos produced per donor per LOPU by season.

**Figure 2 animals-12-03549-f002:**
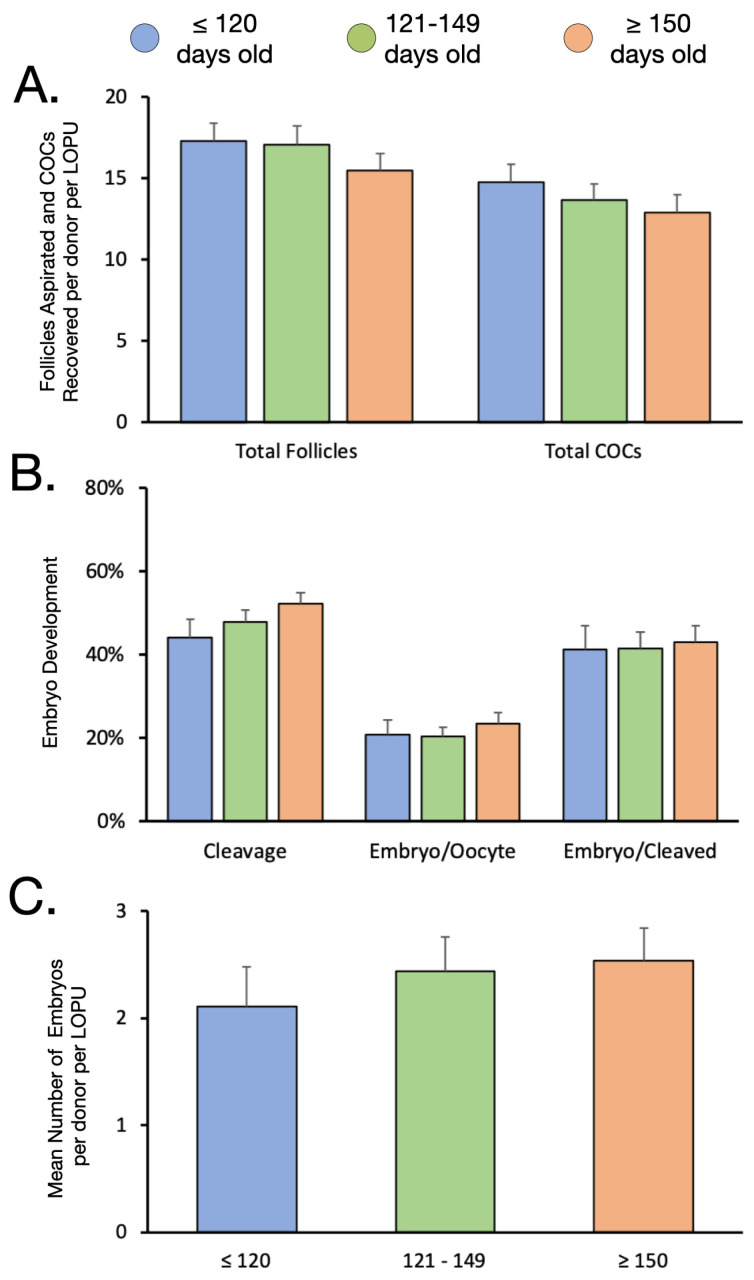
The effect of age on the number of follicles and COCs, embryo development rates and mean number of embryos per donor per LOPU. (**A**): LOPU results according to age showing number of follicles aspirated and number of COCs recovered. (**B**): Embryo development according to age showing cleavage and embryo development rates calculated over oocyte and cleaved embryos. (**C**): The mean number of embryos produced per donor per LOPU according to age category.

**Figure 3 animals-12-03549-f003:**
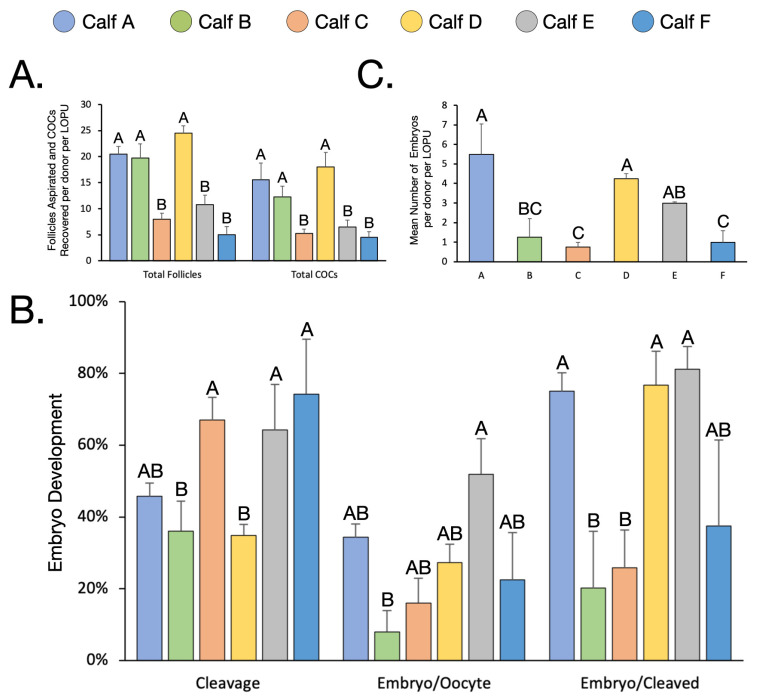
The effect of individual variation. (**A**). The mean number of follicles aspirated and COCs recovered per animal. (**B**). The embryo development rates per animal. (**C**). The mean number of embryos produced per animals per LOPU. Values within the same chart with different script (A–C) differ significantly (*p* < 0.05).

**Figure 4 animals-12-03549-f004:**
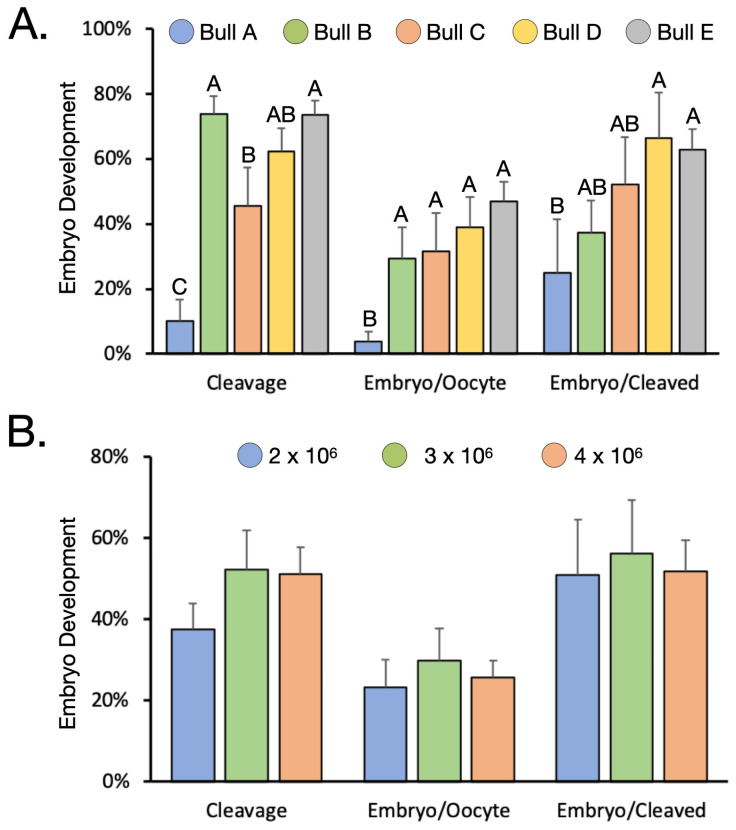
The effect of sire and semen concentration. (**A**). Embryo development according to bull used for IVF. (**B**). The effect of semen concentration for bull D. Values within the same chart with different script (A–C) differ significantly (*p* < 0.05).

**Table 1 animals-12-03549-t001:** Average daily meteorological conditions according to season during the trial period.

Season	Mean T (°C)	Min T (°C)	Max T (°C)	Humidity (%)	Precipitation (mm)	Daylength (h)
Mar–May	7.51 ± 7.2	2.41 ± 6.8	12.60 ± 8.1	63.8 ± 14.8	2.76 ± 6.9	13.55 ± 1.2
Jun–Aug	19.46 ± 3.4	14.51 ± 3.5	24.39 ± 3.7	68.4 ± 12.6	3.39 ± 7.8	15.30 ± 0.7
Sept–Nov	8.34 ± 6.4	3.82 ± 6.1	12.85 ± 7.3	73.3 ± 10.0	2.48 ± 5.0	10.69 ± 1.2
Dec–Feb	−6.07 ± 6.0	−10.92 ± 7.0	−1.22 ± 6.0	73.3 ± 10.5	4.18 ± 7.7	9.80 ± 0.8

Values are expressed as mean ± standard deviation. Data adapted from Environment and Climate Change Canada [50].

**Table 2 animals-12-03549-t002:** Pregnancy rates after embryo transfer according to season.

Season	N	Pregnancies
Mar–May	31	8 (25.8%)
June–Aug	31	7 (22.6%)
Sept–Nov	23	7 (30.4%)
Dec–Feb	33	11 (33.3%)

**Table 3 animals-12-03549-t003:** Embryo Transfer and Pregnancy Establishment.

Variable	Treatment	N	Pregnancies
Number of embryos transferred	1	18	4 (22.2%)
	2	99	29 (29.3%)
Stage of embryo development	Morula	20	4 (20.0%)
	Blastocyst	19	9 (47.4%)
	Hatched	15	4 (26.7%)
Vitrification	Fresh	87	25 (28.7%)
	Vitrified	28	8 (28.6%)

## Data Availability

The data presented in this study is available upon reasonable request from the corresponding author.

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
