# Peer review of "Factors Affecting the Efficiency of In Vitro Embryo Production in Prepubertal Mediterranean Water Buffalo"

_animals, 2022, doi:10.3390/ani12243549_

Round 1

Reviewer 1 Report

Comments:

(1)The delay of sexual maturity is not a problem related to the species but is due to errors due to the breeding technique. In Italy there are now numerous farms that have an age at first calving of 24-28 months.
Journal of Buffalo Science, 2020, 9, 84-91
ISSN: 1927-5196 / E-ISSN: 1927-520X / 20 © 2020 Lifescience Global
Considerations on the Breeding and Weaning of Buffalo Calf

(2)line 57 To further compound this problem, water buffalo mature very slowly compared to cattle, with a delayed onset of puberty and age of sexual maturity, making rates of genetic gain very slow [4].
The discussion of seasonality is commendable.
The authors should point out that although in Canada in spring it is not hot in this country, buffaloes behave like all areas NORTH of the equator and this shows that the light factor is the most important for reproductive purposes. Soc Reprod Fertil Suppl. 2010; 67: 443-55.  doi: 10.7313 / upo9781907284991.034.
Improvement of reproductive performance in domestic dairy water buffalo (Bubalus bubalis)

Author Response

Dear Reviewer 1,

Thank you for taking the time to review our manuscript. We appreciate your feedback and have addressed your comments according to each query below.

(1)The delay of sexual maturity is not a problem related to the species but is due to errors due to the breeding technique. In Italy there are now numerous farms that have an age at first calving of 24-28 months. 
Journal of Buffalo Science, 2020, 9, 84-91 
ISSN: 1927-5196 / E-ISSN: 1927-520X / 20 © 2020 Lifescience Global 
Considerations on the Breeding and Weaning of Buffalo Calf 

R: Thank you for the information and link to the very interesting research article. We acknowledge that puberty and age of sexual maturity are influenced by various factors, with nutrition being a major driver. We have changed the wording of line 57 and added a comment to clarify this on line 59.

(2)line 57 To further compound this problem, water buffalo mature very slowly compared to cattle, with a delayed onset of puberty and age of sexual maturity, making rates of genetic gain very slow [4]. 

R: As mentioned above, we have changed the wording of this sentence, and added a statement that the age at first calving can be affected by managerial factors including nutrition.

The discussion of seasonality is commendable. 

R: Thank you.

The authors should point out that although in Canada in spring it is not hot in this country, buffaloes behave like all areas NORTH of the equator and this shows that the light factor is the most important for reproductive purposes. Soc Reprod Fertil Suppl. 2010; 67: 443-55.  doi: 10.7313 / upo9781907284991.034. 
Improvement of reproductive performance in domestic dairy water buffalo (Bubalus bubalis)

R: Thank you for the important observation. We have added a statement to that effect in the discussion on lines 480-483.

Reviewer 2 Report

The manuscript "Factors affecting the efficiency on in vitro embryo production in prepubertal Mediterranean water buffalo" is well written, scientifically sound, and of interest to the scientific community. It analyses the impact of season, age, and possible individual variations between donor female water buffalos and the sire for in vitro fertilization, finding that laparoscopic ovum pickup and in vitro embryo production can be performed year-round. Furthermore, pregnancy rates are slightly better during autumn and winter and were not affected by embryo vitrification. The research is well justified and these findings are interesting, nonetheless, I would recommend that the references are updated as 48/70 are older than 5 years. Besides this, the manuscript is suitable for publishing.

Author Response

Dear Reviewer 2,

Thank you for taking the time to review our manuscript. We appreciate your feedback and value your comments. Below are our responses to your comments.

The manuscript "Factors affecting the efficiency on in vitro embryo production in prepubertal Mediterranean water buffalo" is well written, scientifically sound, and of interest to the scientific community. It analyses the impact of season, age, and possible individual variations between donor female water buffalos and the sire for in vitro fertilization, finding that laparoscopic ovum pickup and in vitro embryo production can be performed year-round. Furthermore, pregnancy rates are slightly better during autumn and winter and were not affected by embryo vitrification. The research is well justified and these findings are interesting, nonetheless, I would recommend that the references are updated as 48/70 are older than 5 years. Besides this, the manuscript is suitable for publishing.

R: We acknowledge that many of the references are not recent, this is partially because most of our current knowledge regarding oocyte competence in prepubertal animals spans from research conducted in the 90s, before the field was largely abandoned for 20+ years due to  a variety of reasons making the technology uneconomical. It is only with the recent implementation of genomics for herd management that research and interest in the field has taken off again and as a result there is limited recent literature available. However, given the importance of up-to-date information, we have incorporated 7 recent citations (published in the last 4 years) into the manuscript and updated the bibliography accordingly.

Reviewer 3 Report

I belive the manuscript is very interesting, with very good information. My only concern is that the discussion should be largely improved and Language should be reviewed.

Author Response

Dear Reviewer 3,

Thank you for taking the time to review our manuscript. We appreciate your feedback and are happy you found the manuscript interesting. Below are our responses to your comments.

I believe the manuscript is very interesting, with very good information. My only concern is that the discussion should be largely improved and language should be reviewed.

R: Thank you for your feedback, we have updated the discussion and added a new paragraph to discuss the manuscript as a whole (lines 452-461) as well as added some new citations in the bibliography. We have reviewed the language and made grammatical and syntax revisions.